# Effect of Different Vegetable Oils on the Flavor of Fried Green Onion (*Allium fistulosum* L.) Oil

**DOI:** 10.3390/foods12071442

**Published:** 2023-03-29

**Authors:** Ruifang Wang, Lina Qiao, Jing Wang, Junyi Wang, Ning Zhang, Haitao Chen, Jie Sun, Shuqi Wang, Yu Zhang

**Affiliations:** 1Beijing Key Laboratory of Flavor Chemistry, School of Light Industry, Beijing Technology & Business University, Beijing 100048, China; wrf18766537165@163.com (R.W.); theerlking@163.com (L.Q.); wjing2188@163.com (J.W.); cherylwwang@163.com (J.W.); chenht@th.btbu.edu.cn (H.C.); sunjie@btbu.edu.cn (J.S.); wangshuqi@btbu.edu.cn (S.W.); 2College of Food Science, Southwest University, Chongqing 400700, China; zhangyu_512@sina.cn

**Keywords:** green onion (*Allium fistulosum* L.), vegetable oil, flavor, fried oil

## Abstract

The flavor of fried green onion oil (*Allium fistulosum* L.) is widely applied and popular in Chinese cuisine. This work aimed to explore the effects of different varieties of vegetable oils on the flavor profile generation of fried green onion oil. The volatile flavor components of seven different kinds of fried green onion oils, i.e., soybean oil, palm oil, olive oil, corn oil, sunflower oil, camellia oil, and colza oil, were identified and analyzed by sensory analysis, gas chromatography–mass spectrometry (GC-MS) and electronic nose. The results showed that sensory analysis and electronic nose were accepted to detect the odor diversities of different kinds of fried green onion oil sensitively. A total of 103 volatile flavor components were identified positively, and the key aromas included aldehydes and sulfur-containing compounds that correlated highly with green grass, oily, pungent and shallot scent attributes. Meanwhile, fatty acid compositions showed that there were no significant changes in the types of fatty acids before and after frying, but the relative content was not different. Accordingly, the unsaturated fatty acids (UFA, C18:1, C18:2, C18:3, and C20:1) had a significant influence on the flavor of frying oil, which was peculiarly prone to oxidation and heat degradation reactions. These results provided a theoretical basis for further application of fried onion flavor in the food industry.

## 1. Introduction

Fried allium oil has been used extensively in traditional Chinese dishes and has recently enjoyed popularity in the food manufacturing industry [1]. When preparing Chinese dishes, small amounts of seasonings are commonly fried in vegetable oil before adding other ingredients [2]. Conventionally, those technique produces unique flavors at a relatively high frying temperature. Among these allium spices, green onion (*Allium fistulosum* L.) is most commonly used during cooking for Chinese cuisine [3].

During the frying process, desirable properties of foods correlate with color, flavor, texture, and palatability development [4]. Complex physical and chemical interactions between food ingredients and the oil medium can usually promote the formation of special flavors in fried foods [5,6,7]. Additionally, a range of complex chemical reactions occurred between the oil and the food material during the frying process [8,9]. This flavor is mainly determined by the oxidative degradation of fat and the Maillard reaction [10], and the degradation of amino acids also contributed to the generation of flavor compounds at high temperatures [11]. The class of volatile compounds identified from deep-fried oils included alkanes, alcohols, aldehydes, ketones, acids, esters, lactones, and heterocyclic compounds [9,12,13].

Although comprehensive reports on the flavor formation mechanism of green onion fried oil are lacking, volatiles generated from other alliums during heating processes have been studied, and the research showed that major components included sulfides, disulfides, trisulfide, and thiophene [2]. Peng et al. [14] used headspace solid-phase microextraction (HS-SPME) to analyze the volatiles of four fried shallot-flavored oils and indicated that 2-ethyl-3,5-dimethyl-pyrazine, 2,3-dihydro-benzofuran and benzaldehyde contribute to grease flavor while 2-ethyl-3,5-dimethyl-pyrazinemay was associated with shallot scent flavor. Additionally, a Trace GC-MS system equipped with static headspace was adopted to compare the quality of fried shallots oils from microwave heat and gas heat treatment and found that sulfur-containing compounds, including disulfides, trisulfides, and thiophenes, significantly contributed to the flavor of fried shallots [2,15]. Garlic oil processed by the heating process has also been researched. In heat-treated garlic oils, volatiles varied due to the heat treatment or extraction methods used [16,17]. However, most of the detected volatiles were sulfur- or nitrogen-containing compounds.

Herein, the aim of this research was to investigate the effects of different varieties of vegetable oils on the flavor profile generation of fried green onion oil. The fatty acid composition and volatile flavor compounds were determined and compared for seven fried green onion oils using sensory evaluation, GC-MS, and electronic nose. Additionally, the effects of free fatty acid composition on the flavor fingerprint of green onion fried oil were analyzed preliminarily. These results provided a theoretical basis for further application of fried onion flavor in the food industry.

## 2. Materials and Methods

### 2.1. Materials and Chemicals

Fresh native green onion samples (*Allium fistulosum* L.) were purchased from a local market in Beijing, China, and stored in vacuum-packed bags at 4 °C. The vegetable oils, including soybean oil (Yuanbao, first-grade leach pressed), palm oil (Jinhai), olive oil (Olivoila, refined, extra virgin), corn oil (Fulinmen, first-grade press), sunflower oil (Fulinmen, first-grade press), camellia oil (Jinlongyu, first-grade press), and colza oil (Changkang, first-grade press) samples, were purchased from local supermarkets, and the botanical origin and quality of all the samples were guaranteed by the manufacturers.

Chemicals (analytical reagent grade, ≥95% purity) used in this study were as follows: standards of n-alkanes (C_6_~C_30_, ≥99%) were purchased from Sigma-Aldrich Co. (Shanghai, China), 1,2-dichlorobenzene (99%) and dichloromethane (HPLC grade) were purchased from Thermo Fisher Scientific Inc. (Shanghai, China).

### 2.2. Oil Preparation

Fresh green onion stalks were chosen for this experiment. The stalks were cut into sections approximately 1 cm long. A 12~18 cm diameter basin and an induction cooker were used for frying, and an oil thermometer was used to measure the temperature. Through pre-experiment, preparation process of the green onion fried oil was determined to gain a favorable flavor. Generally, vegetable oil (300 g) was added into the basin and was heated to 140 °C. Green onion stalks were then added to the hot oil and fried until the oil reached the desired final temperature of 165 °C. The minimum temperature in this process was slightly lower than 100 °C. The fried stalks and oil were immediately separated once the oil temperature reached 165 °C. Then, the oil was cooled to room temperature as soon as possible. During this procedure, the mixture was stirred continuously to prevent local overheating. The above procedure was repeated twice, and the final deep-fried oil sample was obtained after mixing.

### 2.3. Isolation of the Volatiles by Solvent-Assisted Flavor Evaporation (SAFE)

For the oil samples, fifty grams of oil were mixed with dichloromethane in a ratio of 1:4 (m:v). To this, 1,2-dichlorobenzene was added (50 μL, 0.236 μg/μL) as internal standard and subjected to SAFE analysis to remove non-volatile materials [18]. The procedure was conducted by connecting a glass compact to a distillation vessel to rapidly achieve a high yield of the volatiles from the solvent extracts. A high vacuum was applied to the apparatus to separate the volatiles away from the organic phase [19]. After removal of the non-volatile compounds, the SAFE distillate was dried over anhydrous Na_2_SO_4_ and concentrated to 0.5 mL using a Vigreux column (50 cm × 1 cm, Beijing Jingxing Glassware Co., Ltd., Beijing, China). Samples were prepared in triplicate and stored in 2 mL glass vials at −80 °C for GC-MS analysis.

### 2.4. Gas Chromatography-Mass Spectrometry (GC-MS) Analysis

GC-MS analysis was performed on a TRACE 1310 gas chromatograph (Thermo Fisher Scientific, Waltham, MA, USA) equipped with an ISQ LT mass spectrometer (Thermo Fisher Scientific, Waltham, MA, USA). GCMS-QP2010 (Shimadzu, Japan) was also used. Samples were analyzed on a TG-Wax column (30 m × 0.25 mm i.d., 0.25 µm, Thermo Fisher Scientific, Waltham, MA, USA). Helium was used as the carrier gas at a constant flow rate of 1.0 mL/min. The injector temperature was 250 °C, and the injection volume was 1 µL. The sample was injected in a 1:5 split ratio. The oven temperature was programmed as follows: initial temperature was 40 °C and held for 1 min, then was ramped at 2 °C /min to 130 °C and held for 1 min, and finally ramped at 10 °C/min to 220 °C and held for 8 min. For the MS conditions, the electron ionization mode (EI) was used at 70 eV, the ion source temperature was kept at 250 °C, and the mass range was operated at 35–350 with a solvent delay of 6 min for the samples.

### 2.5. Identification of Volatiles

Volatile compounds from the fried vegetable oils were tentatively identified by GC-MS by comparing their mass spectra with spectra from the NIST 2014 database. The homologous series standards of n-alkanes (C_6_~C_30_) were analyzed under the same chromatographic conditions to calculate the retention index (RI) of the detected compounds using Equation (1) and compare them with the RI in the NIST 2014 database using the same capillary column [20].
(1)$$RI=100\times \left\{n+\left[\mathrm{lg}t^{\prime }(i)-\mathrm{lg}t^{\prime }(n)\right]/\left[\mathrm{lg}t^{\prime }(n+1)-\mathrm{lg}t^{\prime }(n)\right]\right\}$$
where (*n*) and (*n* + 1) represent the number of carbon atoms in the alkanes eluting before and after the compound, respectively, *t*’(*n*) and *t*’(*n* + 1) are the corresponding retention times of the alkanes, and *t*’(*i*) is the retention time of the compound to be identified (*t*’(*n*) < *t*’(*i*) < *t*’(*n* + 1)).

The relative concentrations of the aroma compounds were calculated by relating the peak areas of the volatiles to the peak area of the internal standard (1,2-dichlorobenzene). Content of each compound can be expressed in Equation (2).
(2)$$w(i)=f^{\prime }(i)\times \left[A(i)/A(s)\right]\times w(s)$$
where *w*(*i*) is the detected volatile compound; *w*(*s*) is the concentration of the internal standard; and *A*(*i*) and *A*(*s*) are the peak areas of the volatile compound and the internal standard, respectively, on the GC system. The relative correction factor, *f*’(*i*), was considered to be 1.0 for each component here.

### 2.6. Electronic Nose Analysis

An electronic nose device, PEN 3 E-Nose (Winmuster Airsense Analytic Inc., Schwerin, Germany), was used to analyze the seven kinds of green onion deep-fried oils. The sensor array system consisted of ten metal oxide semiconductors (MOS) of different chemical compositions and thicknesses to provide selectivity towards volatile compound classes, including W1C (aromatic compounds), W5S (broad-range compounds), W3C (ammonia, aromatic compounds), W6S (hydrogen), W5C (aromatic-aliphatic), W1S (methane, broad-range compounds), W1W (sulfur compounds), W2S (broad-alcohol compounds), W2W (sulfur–chlorine), and W3S (methane–aliphatic). Winmuster of the electronic nose was used for data storage and multivariate analysis.

Seven grams of each fried oil was put into 15 mL airtight vials (concentration chamber). The tubes were then heated at 100 °C for 30 min using a SCILOGEX BlueSpin LED digital hot-150 plate magnetic stirrer (MS-H280-Pro, Berlin, CT, USA). Subsequently, a Luer-lock needle (20 gauge) connected to a Teflon tubing (3 mm) was used to perforate the seal (plastic) of the vial to draw the volatile gases in the headspace. Clean air was fed through a second needle connected to a charcoal filter to replace the air sampled from the headspace. During the measurement time (60 s), the sampling unit inhaled the volatile gases present in the headspace at a constant rate causing changes in the sensor’s conductance, which was long enough for the sensor signals to reach a steady value. When the measurement was completed, a standby of 500 s was initiated with the circuit, and the chamber was flushed with clean air until the sensor signals returned to baseline. The E-Nose analysis was performed at least three times for each oil sample.

### 2.7. Determination of the Fatty Acids Contents

Fatty acid methyl esters (FAME) were prepared from oils without frying and the fried oils according to the standard method of GB/T 17376-2008 and GB/T 17377-2008. FAME was analyzed in each sample using heptadecanoic acid as internal standard. The temperature programming and the GC-MS parameters were as previously described [21,22].

### 2.8. Sensory Analysis

A descriptive sensory analysis was carried out in a sensory room with single cubicles at (20 ± 1) °C and relative humidity space. Moreover, the aroma profile analysis was conducted by a panel consisting of ten panelists (6 females and 4 males, aged from 23 to 30 years) from the Beijing Key Laboratory of Flavor Chemistry at Beijing Technology and Business University. They were trained for descriptive analysis and had a lot of experience in sensory profiling of diverse food samples. A 10 g sample was put into a transparent PET bottle (30 mL), and the samples were randomly arranged. The panelists were requested to write down and discuss the odor attributes of fried oil samples. They determined the final list of 7 descriptive terms by consensus. The odor attributes of salty, fried, oily, burnt, cooked vegetables, green grass, and pungent were evaluated. In addition to the descriptive analysis, panelist also scored the aroma intensive using a linear scale ranging from 0 (not perceivable) to 9 (strong perceivable). The final score for each odor attribute was the average of all panelists. Duplicate determinations were performed for each sample.

### 2.9. Statistical Analysis

Statistical analysis was conducted using the XLSTAT software v. 2018 (Addinsoft, New York, NY, USA). Moreover, a one-way analysis of variance (ANOVA) was made to determine differences between samples. Principal component analysis (PCA) was used to study the possible grouping of the fried oil and its corresponding properties. Whilst the partial least squares regression (PLS) was used to show the relationship between aroma sensory variables and volatile compounds considered aroma-contributing substances.

## 3. Results and Discussions

### 3.1. Sensory Analysis of Fried Oils

A descriptive aroma analysis was carried out to obtain the overall flavor of the fried green onion oils. The results can be used to compare the accuracy of the instrumental analyses and sensory evaluations. The strongest flavor profile of the fried oils was the fried note, followed by salty, oily, burnt, cooked vegetables, green grass, and pungent notes. In addition, the seven different frying oils exhibited similar flavor characteristics. Compared to the other six oils, fried soybean oil (S1) had the best flavor profile.

PCA analysis (Figure 1) was applied to the sensory scores. The first two principal components (PC1 and PC2) explained 72.81% of the total variance, while PC1 (47.50%) obtained a much better explanation for the samples than PC2 (25.31%). The PCA results differentiated two groups. The first group was formed by soybean oil (S1), palm oil (S2), and corn oil (S4) and distributed on the positive side of PC1. These fried oils were correlated to the sensory attributes of salty, fried, cooked vegetables, pungent, and green grass. The second group was formed by olive oil (S3), sunflower oil (S5), camellia oil (S6), and colza oil (S7) and located on the negative side of PC1. They were related to the sensory attributes of oily and burnt. In general, the sensory evaluation of the fried oil presented that strong roasted and salty notes of flavor characteristics could be created by frying, and a significant role was played by the sensory evaluation in contrasting the differences in flavor.

### 3.2. Volatile Compound Analysis of Fried Oils

The volatile compounds identified in the seven fried oils are given in Table 1. A total of 103 volatiles were identified and included 2 alkanes, 18 aldehydes, 16 alcohols, 12 ketones, 5 acids, 5 esters, 19 furans and furanones, 19 sulfur-containing compounds, and 7 nitrogen-containing compounds. This indicated that the frying process had a significant effect on the flavor characteristics of fried green onion oils.

Figure 2 is a heat map showing the concentrations of the different volatiles identified in the seven fried oils. Furan and furanone compounds were the highest concentrations in the seven oils, followed by aldehydes and sulfur-containing compounds. Clearly, the content of volatile flavor compounds in the colza fried oil (S7) was higher than the others. Combined with Table 1, the results showed that the most abundant aroma compounds in the fried oils were (*E*)-2-heptenal (A30), dimethyl trisulfide (A34), furfural (A44), pyranone (A93), and 5-hydroxymethylfurfural (A96). Additionally, aldehydes and sulfur-containing compounds were also identified with the higher concentrations in the green onion fried oils. Thereinto, (*E*)-2-hexenal (A18), (*E*,*E*)-2,4-heptadienal (A46), dimethyl disulfide (A1), and dimethyl trisulfide (A34) were the most prominent compounds in the green onion fried in soybean oil (S1), and 5-methyl-2-furancarboxaldehyde (A61), nonanal (A35), methyl 1-propenyl disulfide (A26), and (*E*)-1-methyl-3-(prop-1-en-1-yl) trisulfane (A63) were the most prominent compounds in the extracts of the green onion fried in palm oil (S2), and (*E*,*E*)-2,4-decadienal (A76), (*E*)-2-nonenal (A53), and 2-undecenal (A71) were the most primary compounds in the green onion fried in camellia oil (S6).

### 3.3. Electronic Nose Analysis of the Fried Oils

The correlations between the chemical classes of volatile compounds and the intensities of the electronic nose were analyzed by PLS [23,24], and the results showed in Figure 3. The results indicate that most of the X variables (relative abundance of the chemical classes of volatile compounds) and Y variables (intensities of the electronic nose) are located within the ellipse (*r*^2^ = 100%, *r*^2^ represents the degree of interpretation) [25]. A reliable model (*Q*^2^ = 0.89 ≥ 0.50) was devised for the intensities of the electronic nose and the species of fried oil using all the volatile compounds data [26]. The plot (explaining 74.0% of the total variance) suggested a correlation between the chemical classes of volatile compounds and the intensities of the electronic nose and their species of the fried oils analyzed. Moreover, the results (Figure 3) showed that palm oil (S2), corn oil (S4), and sunflower oil (S5) were positioned in the first quadrant; camellia oil (S6) was placed in the second quadrant; and olive oil (S3) and colza oil (S7) were distributed in the third quadrant while soybean oil (S1) was located in the fourth quadrant.

Additionally, chemical groups of volatile compounds and electronic nose sensors were highly correlated and associated with the fried oils. In terms of the electronic nose sensors, Figure 3 showed positive correlations between fried oil samples and the sensors. In general, sensor 1 (W1C, aromatic compounds), sensor 3 (W3C, ammonia compounds), and sensor 5 (W5C, aromatic–aliphatic) were related to palm oil (S2) and corn oil (S4); sensor 10 (W3S, methane–aliphatic), sensor 4 (W6S, hydrogen), sensor 8 (W2S, broad-alcohol compounds), and sensor 6 (W1S, methane compounds) were correlated to olive oil (S3) and colza oil (S7). Likewise, sensor 2 (W5S, broad-range compounds), sensor 7 (W1W, sulfur compounds), and sensor 9 (W2W, sulfur–chlorine) were closely associated with soybean oil (S1), which agreed with the volatile compound analysis and sensory evaluation results.

In addition, the flavor fingerprints could be established to differentiate the different kinds of fried green onion oils by the electronic nose and GC-MS analysis results. For example, the most important attributes of the flavor of soybean oil (S1) were the sulfur-containing components and aldehydes, including 3,4-dimethyl-thiophene (A24), dimethyl trisulfide (A34), methyl propyl disulfide (A20), and (*E*)-2-heptenal (A30), etc. The noteworthy components of the flavor of colza oil (S7) were aldehydes, furan, and furanones, including (*E*,*E*)-2,4-heptadienal (A46), 5-hydroxymethylfurfural (A96), furyl hydroxymethyl ketone (A87), and 3,5-dihydroxy-2-methyl-4H-pyran-4-one (A94). Generally, the electronic nose was effective for the executant to differentiate the different kinds of fried oil or authenticity.

### 3.4. Fatty Acid Compositions of Fried Oils

Figure 4 shows the changes in the saturated fatty acids (SFA), unsaturated fatty acids (UFA), monounsaturated fatty acids (MUFA), and polyunsaturated fatty acids (PUFA) contents in the seven fried oils.

The fried oils whose contents of SFA increased after frying included soybean oil, palm oil, olive oil, corn oil, and camellia oil, while sunflower oil and colza oil decreased. For the UFA, the contents of soybean oil, olive oil, corn oil, and camellia oil were increased after frying, and the palm oil, sunflower oil, and colza oil were clearly decreased. Specifically, the fried olive oil had the most increase in the UFA content. For the MUFA and PUFA, the contents of MUFA in soybean oil, olive oil, corn oil, and camellia oil were increased after frying. Moreover, the contents of PUFA slightly increased, including soybean oil, palm oil, olive oil, corn oil, and camellia oil. Conversely, MUFA and PUFA decreased for sunflower oil and colza oil. Significantly, the reason the total amount of UFA of the fried palm oil decreased was that the decrease of MUFA was significantly higher than the increase of PUFA after frying.

Contrast analysis showed that while frying, the composition of fatty acids in oils influenced the flavor of fried oil. Among them, UFA (such as oleic acid, linoleic acid, and arachidonic acid) were prone to oxidation due to their double bonds resulting in the formation of peroxides [27] further decomposed to volatile carbonyl compounds such as ketones and aldehydes, acids, contributing to the characteristic flavor profile [28]. Fatty acids containing hydroxyl groups are dehydrated and cycled to form lactones with pleasant aromas [29]. Additionally, products from thermal degradation continue to react with proteins and amino acids in the matrix to obtain heterocyclic compounds with special aromas [30].

## 4. Conclusions

Generally, the effect of different varieties of vegetable oils on the flavor profile generation of fried green onion oil was determined and differentiated. Likewise, the correlation between the composition of fatty acid and the specific flavor of the fried green onion oils was discussed. The electronic nose was sensitive to detect the subtle differences in the aroma of seven vegetables after frying green onions. A total of 103 volatile compounds were identified, including alkanes, aldehydes, alcohols, ketones, acids, esters, furan and furanones, sulfur-containing compounds, and nitrogen-containing compounds; some of them, such as (**E*,*E**)-2,4-decadienal, (*E*)-2-heptenal, hexanal, dimethyl trisulfide, and methyl propenyl disulfide, were highly correlated with the flavor characteristic of the fried oils. There was an obvious correlation between the electronic nose sensor data and the key aroma compounds, and it was feasible to establish the flavor fingerprints of different kinds of green onion fried oils. The fatty acid composition and relative content of fried green onion oils were closely related to the types of vegetable oil. MUFA and PUFA, such as C18:1, C18:2, C18:3, and C20:1, changed significantly during frying and were susceptible to oxidation and thermal degradation reactions. It had a significant effect on the flavor of fried green onion oils. This work provides preliminary data for future research to probe the factors influencing the flavor and quality of fried green onion oil products during the frying process.

## Figures and Tables

**Figure 1 foods-12-01442-f001:**
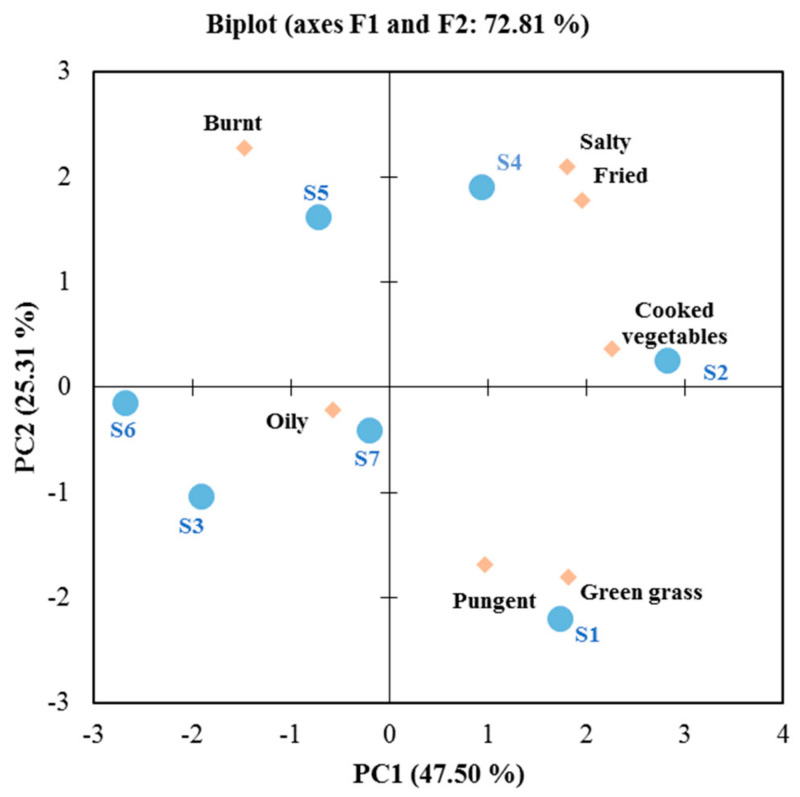
Principal component analysis (PCA) of seven fried green onion oils on the sensory evaluations. (S1: soybean oil; S2: palm oil; S3: olive oil; S4: corn oil; S5: sunflower oil; S6: camellia oil; S7: colza oil.).

**Figure 2 foods-12-01442-f002:**
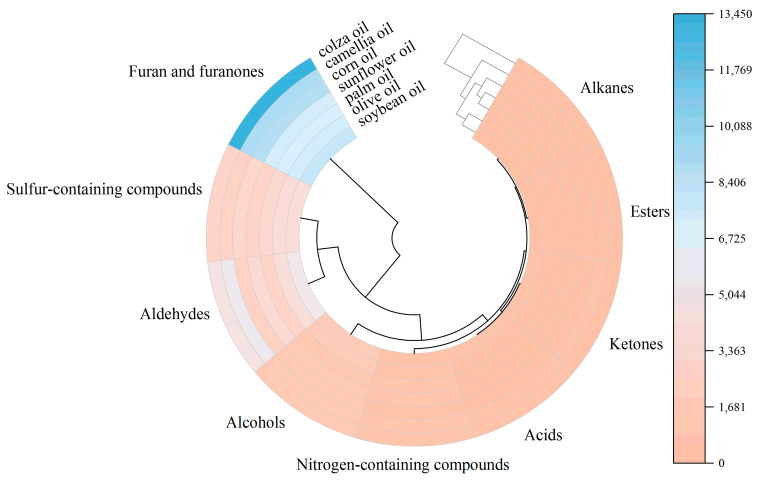
Heat map analysis of volatile flavor compounds in the seven fried green onion oils.

**Figure 3 foods-12-01442-f003:**
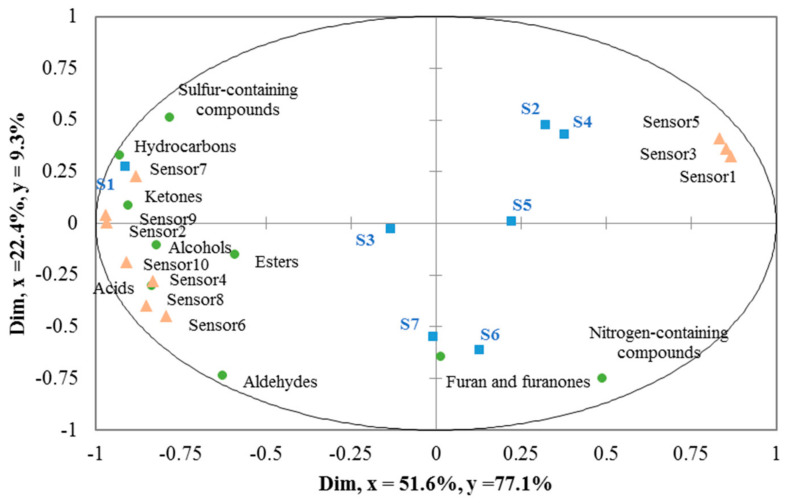
Partial Least Square (PLS) explaining the relationship between chemical groups of volatile compounds and electronic nose sensors in the green onion deep-fried oils. (S1: soybean oil, S2: palm oil, S3: olive oil, S4: corn oil, S5: sunflower oil, S6: camellia oil, S7: colza oil; Sensor 1~10 pointed to sensors of W1C, W5S, W3C, W6S, W5C, W1S, W1W, W2S, W2W, and W3S consequently).

**Figure 4 foods-12-01442-f004:**
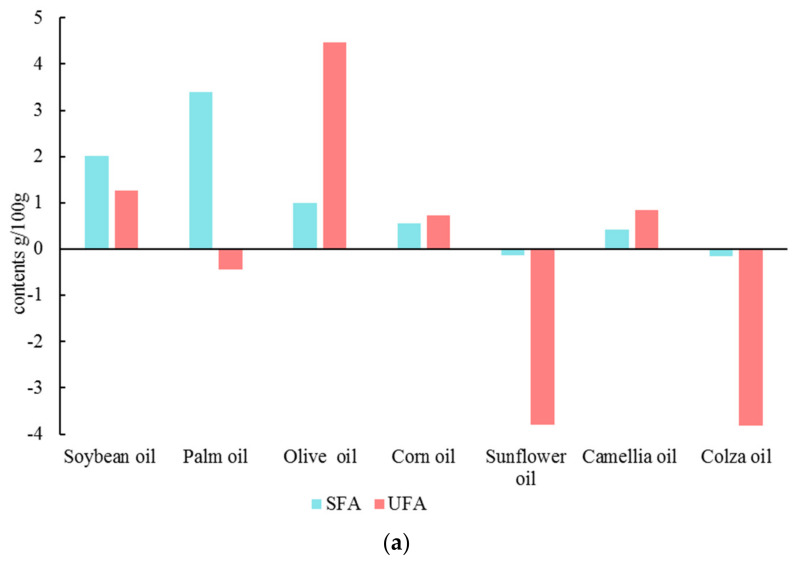
Changes of fatty acids in deep-fried oils after frying. ((**a**): variations of saturated fatty acid (SFA) and unsaturated fatty acid (UFA); (**b**): variations of monounsaturated fatty acid (MUFA) and polyunsaturated fatty acid (PUFA)).

**Table 1 foods-12-01442-t001:** Volatile compounds detected in seven fried green onion oils and their concentrations.

No.	Compound	Concentration (μg/kg) ^1^							RI ^2^ (Exp/Lit)
		Soybean Oil	Palm Oil	Olive Oil	Corn Oil	Sunflower Oil	Camellia Oil	Colza Oil	
A1	Dimethyl disulfide	241.99 ± 47.52	107.96 ± 4.65	139.04 ± 22.07	78.58 ± 2.34	66.73 ± 6.03	63.52 ± 7.86	51.52 ± 11.67	1080/1071
A2	Hexanal	697.71 ± 96.14	97.87 ± 0.95	436.75 ± 80.36	115.33 ± 6.94	265.05 ± 24.58	173.89 ± 22.65	64.21 ± 10.27	1090/1097
A3	2-Methyl-2-butenal	31.43 ± 4.46	17.39 ± 2.93	32.01 ± 7.76	28.69 ± 2.87	21.76 ± 3.97	17.38 ± 2.16	14.01 ± 2.42	1102/1093
A4	4-Pentenal	nd ^3^	nd	nd	nd	nd	nd	8.83 ± 1.24	1114/1123
A5	3-Methyl-thiophene	10.16 ± 2.08	4.01 ± 0.21	7.19 ± 1.44	4.21 ± 0.09	21.76 ± 3.97	3.51 ± 0.49	5.72 ± 1.83	1125/1123
A6	(*E*)-2-Pentenal	60.89 ± 8.61	6.32 ± 0.47	27.22 ± 8.99	5.68 ± 0.14	3.83 ± 0.47	8.22 ± 1.33	34.61 ± 4.38	1134/1128
A7	Pentyl-oxirane	12.64 ± 1.31	nd	nd	nd	nd	nd	nd	1146/1153
A8	1-Penten-3-ol	95.75 ± 17.48	11.6 ± 1.19	64.64 ± 2.33	9.96 ± 0.18	8.49 ± 0.92	15.3 ± 1.14	35.71 ± 7.46	1162/1176
A9	β-Myrcene	9.31 ± 0.9	nd	nd	nd	nd	nd	nd	1167/1169
A10	3-Penten-2-ol	803.38 ± 149.17	576.45 ± 41.85	923.16 ± 169.27	708.55 ± 49.59	687.3 ± 92.12	540.74 ± 48.68	583.58 ± 165.59	1174/1178
A11	2-Heptanone	44.59 ± 3.89	nd	nd	nd	nd	nd	nd	1186/1187
A12	Heptanal	48.89 ± 4.23	18.28 ± 0.32	65.82 ± 7.38	12.59 ± 0.92	22.63 ± 1.33	54.94 ± 3.97	21.59 ± 1.62	1189/1188
A13	2,4-Dimethyl-thiophene	18.63 ± 3.28	10.6 ± 0.68	18.71 ± 3.29	12.44 ± 0.57	14.78 ± 2.86	9.75 ± 0.88	8.83 ± 1.81	1194/1197
A14	Limonene	69.36 ± 5.8	14.7 ± 0.19	23.37 ± 3.26	7.7 ± 0.51	6.91 ± 0.21	9.01 ± 0.56	5.34 ± 0.29	1199/1209
A15	3-Methyl-2-butenal	62.24 ± 11.58	50.03 ± 17.49	67.03 ± 11.17	42.62 ± 3.91	36.78 ± 2.9	38.95 ± 8.42	35.78 ± 4.6	1205/1202
A16	2,3-Dimethyl-2-butanol	45.49 ± 9.29	38.41 ± 5.6	66.95 ± 15.6	36.48 ± 3.53	37.36 ± 5.89	35.06 ± 8.07	40.62 ± 6.82	1214/—
A17	3-Hexen-2-one	7.32 ± 1.49	nd	nd	nd	nd	nd	6.4 ± 1.02	1218/1209
A18	(*E*)-2-Hexenal	94.49 ± 12.99	18.49 ± 0.65	48.75 ± 9.92	29.48 ± 0.36	41.3 ± 4.21	30.18 ± 3.25	24.42 ± 3.4	1222/1218
A19	4-Octanone	nd	nd	nd	nd	nd	2.75 ± 0.28	nd	1228/1224
A20	Methyl propyl disulfide	49.86 ± 7.97	44.88 ± 4.04	52.24 ± 6.38	28.29 ± 1.05	33.96 ± 4.74	33.13 ± 5.72	25.65 ± 4.04	1232/1242
A21	2-Pentyl-furan	108.33 ± 6.03	25.31 ± 1.22	82.88 ± 10.17	33.87 ± 1.25	65.28 ± 6.65	40.22 ± 4.8	15.59 ± 1.89	1234/1233
A22	3-Hydroxy-3-methyl-2-butanone	29.58 ± 3.17	19.49 ± 1.7	34.49 ± 7.09	19.68 ± 0.55	22.52 ± 2.16	18.93 ± 1.79	20.09 ± 3.71	1244/1247
A23	1-Pentanol	57.44 ± 5.01	21.36 ± 1.43	36.74 ± 2.73	12.96 ± 0.86	32.48 ± 2.35	62.54 ± 3.59	13.13 ± 0.84	1252/1252
A24	3,4-Dimethyl-thiophene	246.79 ± 28.99	153.56 ± 10.09	241.91 ± 29.62	172.47 ± 10.74	187.96 ± 13.1	155.25 ± 11.82	146.26 ± 19.08	1254/1250
A25	(E)-1-Methyl-2-(prop-1-en-1-yl)disulfane	247.81 ± 31.82	145.04 ± 6.56	206.8 ± 31.53	129.3 ± 2.77	119.86 ± 9.58	94.65 ± 11.9	98.61 ± 15.08	1264/—
A26	Methyl 1-propenyl disulfide	983.84 ± 131.68	687.34 ± 21.95	899.81 ± 118.97	650.15 ± 21.91	661.94 ± 54.55	514.91 ± 46.92	525.01 ± 50.43	1290/1274
A27	Octanal	37.25 ± 1.49	18.31 ± 5.41	68.17 ± 2.66	9.09 ± 3.51	8.31 ± 2.86	79.72 ± 1.63	21.33 ± 1.1	1292/1291
A28	2-Heptanol	60.79 ± 6.61	62.03 ± 6.14	77.07 ± 13.76	52.23 ± 2.48	61.4 ± 2.73	47.86 ± 2.43	48.44 ± 8.15	1311/1286
A29	Prenol	23.94 ± 2.7	19.14 ± 0.88	32.7 ± 5.37	21.42 ± 0.32	20.99 ± 2.34	20.15 ± 3.61	22.88 ± 2.46	1324/1328
A30	(*E*)-2-Heptenal	1275.27 ± 112.83	148.22 ± 10.43	388.62 ± 31.43	295.99 ± 8.47	570.6 ± 23.15	360.89 ± 18.07	210.45 ± 13.95	1326/1317
A31	2,3-Octanedione	209.26 ± 6	nd	nd	nd	nd	17.46 ± 1.87	nd	1330/1336
A32	1,4-Pentadien-3-ol	15.1 ± 0.88	20.03 ± 0.74	25.12 ± 1.99	17.55 ± 1.45	17.23 ± 1.98	19.18 ± 1.85	21.31 ± 3.85	1340/—
A33	2-Cyclopenten-1-one	29.88 ± 2.53	19 ± 2.05	32.08 ± 1.67	21.9 ± 1.6	23.72 ± 0.97	23.2 ± 1.04	25.74 ± 3.54	1355/1349
A34	Dimethyl trisulfide	983.34 ± 86.74	790.9 ± 7.05	855.51 ± 93.78	640.16 ± 15.39	628.49 ± 31.57	569.62 ± 37.54	511.38 ± 33.31	1377/1369
A35	Nonanal	159.07 ± 13.95	294.15 ± 27.81	634.53 ± 25.84	101.19 ± 10.28	124.88 ± 3.31	833 ± 36.99	373.73 ± 79.14	1395/1379
A36	Isoxazole	58.39 ± 6.19	28.21 ± 2.97	39.73 ± 4.15	27.95 ± 2.48	29.72 ± 1.55	13.89 ± 1.94	35.72 ± 4.96	1401/—
A37	Propyl sulfide	174.8 ± 24.33	185.82 ± 2.09	252.26 ± 30.53	156.88 ± 7.22	121.84 ± 12.52	48.58 ± 15.3	124.59 ± 7.25	1408/1069
A38	(*E*)-2-Octenal	205.92 ± 6.35	66.32 ± 2.37	179.61 ± 5.98	66.63 ± 5.32	110.59 ± 3.39	192.08 ± 9.49	77.49 ± 11.81	1430/1426
A39	1-Allyl-2-isopropyldisulfane	75.01 ± 0.67	65.68 ± 1.54	76.56 ± 8.64	51.45 ± 3.21	58.96 ± 2.23	71.14 ± 1.98	46.6 ± 2.77	1435/—
A40	1-Octen-3-ol	238.57 ± 1.02	43.34 ± 1.57	80.36 ± 0.95	84.81 ± 6.38	166.37 ± 4.79	103.28 ± 6.47	85.24 ± 14.31	1454/1476
A41	Methional	25.65 ± 6.12	nd	40.43 ± 12.6	11.82 ± 3.97	13.82 ± 2.43	nd	nd	1457/1474
A42	1-Heptanol	nd	nd	nd	nd	nd	38.75 ± 7.5	nd	1457/1467
A43	Acetic acid	101.22 ± 16.23	70.35 ± 7.87	nd	50.46 ± 24.39	41.52 ± 19.6	88.57 ± 12.01	73.22 ± 47.03	1464/1484
A44	Furfural	1897.96 ± 249.09	1103.1 ± 28.87	1685.38 ± 314.83	1029.43 ± 42.3	1197.84 ± 112.87	1099.88 ± 74.34	1528.11 ± 164.48	1468/1471
A45	2-Ethyl-1-hexanol	13.83 ± 0.19	14.5 ± 4.67	14.64 ± 2.44	12.24 ± 0.47	15.09 ± 0.65	13.89 ± 2.22	16.69 ± 1.43	1492/1494
A46	(*E*,*E*)-2,4-Heptadienal	578.21 ± 10.94	54.57 ± 1.28	191.56 ± 4.32	46.94 ± 4.1	8.83 ± 0.33	151.51 ± 7.45	1044.81 ± 167.79	1495/1490
A47	1-(Ethylthio)-2-methyl-1-propene	31.43 ± 4.05	36.1 ± 0.97	34.12 ± 7.31	35.13 ± 2.26	24.26 ± 2.86	12.37 ± 6.85	22.05 ± 1.37	1498/—
A48	1-(2-Furanyl)-ethanone	203.08 ± 17.33	196.44 ± 2.58	245.85 ± 22.63	147.46 ± 8.5	183.15 ± 6.2	203.22 ± 8.83	179.65 ± 3.74	1506/1512
A49	3-Nonen-2-one	32.16 ± 4.99	26.27 ± 3.16	30.12 ± 4.86	17.33 ± 3.59	28.88 ± 3.02	45.56 ± 4.15	25.07 ± 8.77	1512/1508
A50	3-Methoxy-1-butanol	39.71 ± 1.55	38.98 ± 0.99	35.98 ± 13.39	39.78 ± 1.91	36.63 ± 6.25	42.34 ± 4.98	71.46 ± 14.85	1518/—
A51	3,5-Octadien-2-one	nd	nd	nd	nd	nd	nd	9.66 ± 2.51	1524/1531
A52	Methyl propyl trisulfide	77.71 ± 4.3	92.75 ± 3.08	94.03 ± 5.56	68.61 ± 5.64	13.08 ± 1.98	71.21 ± 5.92	90.66 ± 5.14	1523/1529
A53	(*E*)-2-Nonenal	19.56 ± 2.16	15.18 ± 0.42	39.35 ± 5.84	12.94 ± 1.47	22.81 ± 3.11	127.67 ± 10.97	51.48 ± 17.62	1536/1535
A54	4-Ethylcyclohexanol	74.89 ± 17.22	nd	nd	35.27 ± 6.09	75.22 ± 10.22	42.24 ± 3.73	39.15 ± 6.54	1539/—
A55	2,4-Dimethyl-cyclohexanol	82.07 ± 3.02	10.76 ± 0.68	47.42 ± 4.5	25.98 ± 2.39	48.35 ± 3.7	26.63 ± 4.23	23.2 ± 0.33	1542/—
A56	Linalool	18.63 ± 2.61	nd	nd	nd	nd	nd	nd	1551/1554
A57	2-Pyridinecarboxaldehyde	nd	11.88 ± 3.59	16.3 ± 3.63	6.68 ± 2.94	nd	nd	6.13 ± 1.13	1551/1470
A58	Linalyl acetate	51.95 ± 13.89	nd	nd	nd	nd	nd	nd	1559/1556
A59	1-Octanol	nd	nd	63.64 ± 4.5	nd	nd	79.52 ± 8.74	36.34 ± 11.58	1559/1540
A60	Dimethyl sulfoxide	79.37 ± 2.23	61.53 ± 0.94	63.77 ± 19.7	62.15 ± 10.12	56.56 ± 3.97	68.65 ± 4.8	125.8 ± 40.41	1569/1620
A61	5-Methyl-2-furancarboxaldehyde	433.16 ± 16.5	601.08 ± 17.05	587.71 ± 41.34	370.12 ± 29.01	473.5 ± 5.92	644.35 ± 22.33	477.68 ± 48.77	1576/1610
A62	4-Cyclopentene-1,3-dione	73.67 ± 0.83	70.67 ± 14.38	63.34 ± 16.59	56.02 ± 3.61	62.54 ± 3.91	56.2 ± 13.29	nd	1587/1573
A63	(E)-1-Methyl-3-(prop-1-en-1-yl)trisulfane	621.9 ± 72.03	634.74 ± 40.89	462.56 ± 46.64	529.91 ± 74.19	532.18 ± 25.16	497.15 ± 28.12	723.03 ± 114.38	1596/1588.7
A64	Butyrolactone	15.58 ± 2.23	30.38 ± 1.78	28.42 ± 2.15	21.24 ± 3.08	25.15 ± 1.61	18.58 ± 0.36	nd	1628/1626
A65	Benzeneacetaldehyde	642.02 ± 34.9	892.03 ± 56.84	920.11 ± 45.81	948.68 ± 62.97	624.12 ± 15.97	1030.56 ± 79.47	1199.87 ± 235.09	1643/1638
A66	2-Furanmethanol	415.36 ± 40.24	376.09 ± 15.64	423.39 ± 51.9	311.61 ± 17.46	370.04 ± 26.66	416.17 ± 31.23	313.78 ± 14.87	1665/1649
A67	3-Methyl-1H-pyrazole	nd	nd	nd	nd	nd	17.73 ± 2.12	nd	1683/1690.3
A68	5-Ethyldihydro-2(3H)-furanone	71.39 ± 16.12	nd	nd	nd	nd	nd	144.94 ± 35.41	1697/1687
A69	(*E*,*E*)-2,4-Nonadienal	14.83 ± 3.23	5.48 ± 0.36	23.6 ± 2.6	10.94 ± 3.25	17.28 ± 0.92	39.01 ± 6.49	10.4 ± 3.53	1704/1703
A70	5-Methyl-2-furanmethanol	121.24 ± 20.22	95.5 ± 9.32	123.03 ± 44.15	91.08 ± 8.19	110.71 ± 22.4	74.5 ± 14.35	42.79 ± 4.94	1725/1722
A71	2-Undecenal	57.82 ± 3.37	79.14 ± 1.81	131.86 ± 8.07	53.12 ± 10.62	53.82 ± 3.83	277.22 ± 37.15	145.22 ± 47.28	1752/1736
A72	2,4-Decadienal	414.19 ± 50.89	320.12 ± 16.75	324.24 ± 40.1	203.17 ± 24.76	567.46 ± 25.52	770 ± 94.11	440.71 ± 157.85	1766/1767
A73	2-Hydroxy-2-cyclopenten-1-one	46.56 ± 2.16	55.86 ± 5.63	51.15 ± 3.22	47.9 ± 4.3	37.54 ± 0.2	58.74 ± 2.22	56.52 ± 11.02	1772/1769
A74	1-(2-Butoxyethoxy)-ethanol	108.94 ± 15.84	95.49 ± 28.26	192.47 ± 62.22	95.28 ± 9.24	73.22 ± 9.9	76.97 ± 14.56	299.47 ± 194.71	1791/1800
A75	1,3-Butadiene-1-carboxylic acid	nd	nd	nd	nd	35.44 ± 1.46	nd	52.49 ± 14.58	1805/1879.0
A76	(*E*,*E*)-2,4-Decadienal	750.94 ± 79.66	589.93 ± 34.1	600.26 ± 73.53	433.43 ± 46.73	1002.09 ± 25.2	1414.77 ± 148.71	897.69 ± 313.2	1810/1807
A77	3-Methyl-1,2-cyclopentanedione	18.03 ± 0.91	29 ± 2.4	21.01 ± 2.96	20.2 ± 1.24	17.98 ± 0.6	29.73 ± 3.19	34.07 ± 13.21	1829/1800.1
A78	2-(Ethylthio)-ethanol	89.1 ± 2.83	79.25 ± 3.51	83.02 ± 5.22	70.57 ± 10.08	63.59 ± 0.74	65.88 ± 5.07	89.02 ± 20.11	1834/—
A79	5,6-Dihydro-2H-pyran-2-one	12.78 ± 2.77	14.71 ± 1.22	19.83 ± 4.88	11.44 ± 2.92	11.78 ± 3	25.38 ± 6.9	29.35 ± 8.18	1852/1838
A80	3-(2-Furanyl)-2-propenal	nd	nd	nd	nd	nd	7.71 ± 2.44	nd	1858/1851
A81	Hexanoic acid	293.68 ± 40.49	71.91 ± 11.63	143.26 ± 14.63	41.85 ± 9.57	33.22 ± 2.58	59.55 ± 2.49	33.06 ± 12.67	1861/1854
A82	Dihydro-5-pentyl-2(3H)-furanone	nd	nd	nd	nd	nd	11 ± 1.23	nd	1912/2011
A83	2-Propenamide	15.44 ± 2.74	18.09 ± 1.47	nd	11.82 ± 0.12	10.55 ± 0.91	13.49 ± 0.56	23.93 ± 6.66	1940/1943
A84	1-(1H-Pyrrol-2-yl)-ethanone	274 ± 39.15	366.65 ± 11.97	338.54 ± 23.55	299.06 ± 43.52	409.82 ± 119.15	421.81 ± 59.14	439.08 ± 140.43	1970/1969
A85	S-Methyl methanethiosulphonate	111.4 ± 36.92	89.18 ± 15.52	69.05 ± 13.06	52.5 ± 14.43	63 ± 6.01	88.45 ± 16.5	105.74 ± 27.57	1976/—
A86	2,5-Furandicarboxaldehyde	33.42 ± 7.46	40.55 ± 3.95	42.24 ± 4.32	32.8 ± 7.22	32.68 ± 4.5	92.04 ± 16.56	68.63 ± 20.79	1982/1982.7
A87	Furyl hydroxymethyl ketone	270.79 ± 36.17	408.53 ± 23	346.17 ± 35.71	289.25 ± 48.46	316.33 ± 19.01	542.79 ± 82.05	616.99 ± 213.62	2002/1989
A88	1H-Pyrrole-2-carboxaldehyde	16.16 ± 3.16	23.96 ± 2.1	20.87 ± 3.19	20.6 ± 4.18	21.47 ± 6.31	24.49 ± 3.54	29.49 ± 5.83	2023/2032
A89	Furaneol	21.49 ± 5.95	21.23 ± 2.85	13.01 ± 4.27	24.92 ± 3.63	26.83 ± 4.07	23.7 ± 5.95	19.58 ± 5.57	2033/2031
A90	5-Methyl-1H-pyrrole-2-carboxaldehyde	28.01 ± 4.03	43.89 ± 3.56	37.79 ± 3.12	29.84 ± 2.42	36.15 ± 2.35	43.47 ± 15.52	52.37 ± 22.13	2104/2088.1
A91	Nonanoic acid	nd	nd	nd	nd	36.44 ± 13.5	107.34 ± 23.89	36.19 ± 9.05	2174/2172
A92	5-Acetoxymethyl-2-furaldehyde	11.84 ± 5.34	15.48 ± 5.61	24.15 ± 8.84	13.6 ± 3.83	13.52 ± 2.54	30.41 ± 7.85	30.05 ± 9.29	2196/2194
A93	Pyranone	1353.16 ± 353.41	1175.05 ± 144.23	830 ± 167.92	1334.23 ± 86.09	1006.09 ± 517.03	1066.88 ± 220.03	1110.83 ± 342.4	2264/2267
A94	3,5-Dihydroxy-2-methyl-4H-pyran-4-one	211.34 ± 87.82	310.31 ± 48.35	92.78 ± 88.34	1684.56 ± 258.29	751.88 ± 267.35	1461.29 ± 314.68	2468.79 ± 958.78	2305/2309
A95	(S)-( + )-2’,3’-Dideoxyribonolactone	33.7 ± 7.05	50.23 ± 5.99	30.74 ± 14.35	43.73 ± 5.9	42.96 ± 5.2	74.15 ± 11.23	79.78 ± 31.09	2475/—
A96	5-Hydroxymethylfurfural	2824.54 ± 118.56	2782.52 ± 170.58	2823.73 ± 236.2	3296.32 ± 420.13	2972.4 ± 185.56	3305.51 ± 293.79	6344.05 ± 2309.27	2502/2512
A97	1,2-Benzenedicarboxylic acid, bis(2-methylpropyl) ester	39.38 ± 22.62	22.29 ± 2.66	30.85 ± 8.08	26.32 ± 2.31	27.73 ± 3.57	19.35 ± 0.98	59.37 ± 25.28	2539/2526
A98	3-Thiopheneethanol	43.84 ± 9.17	44.94 ± 3.54	25.56 ± 9.65	55.29 ± 6.34	49.18 ± 4.31	42.76 ± 10.59	56.9 ± 15.79	2577/—
A99	Dihydro-4-hydroxy-2(3H)-furanone	nd	25.8 ± 5.27	nd	20.95 ± 0.93	17.11 ± 1.28	38.16 ± 1.76	43.21 ± 8.27	2596/—
A100	Benzyl benzoate	21.51 ± 3.88	20.29 ± 0.59	23.25 ± 6.42	25.03 ± 5.15	24.65 ± 7.1	16.98 ± 2.67	29.3 ± 10.2	2620/2624.7
A101	Dibutyl phthalate	36.17 ± 8.49	33.72 ± 3.04	45.04 ± 7.54	40.68 ± 3.76	41.91 ± 10.97	32.92 ± 1.95	97.33 ± 51.62	2692/2680
A102	Ethyl N-(o-anisyl)formimidate	72.77 ± 16.75	134.66 ± 19.75	119.68 ± 34.37	97.33 ± 13.96	92.36 ± 9.33	156.85 ± 24.81	146.48 ± 61.29	2754/—
A103	n-Hexadecanoic acid	56.63 ± 14.17	100.85 ± 35.67	81.9 ± 19.06	61.39 ± 12.02	103.79 ± 33.98	63.33 ± 6.64	137.04 ± 45.4	>2900/2906

^1^ The concentrations of compounds were calculated using the internal standard method, and the results are shown as the Mean ± standard deviation. ^2^ The retention indices of compounds on the TG-Wax column were calculated against the GC-MS retention time of n-alkanes (C6–C30). “Exp”: experimentally measured on the TG-Wax. “Lit”: published retention index. ^3^ nd: non-detected.

## Data Availability

The data are available from the corresponding author.

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
