# Peer review of "Effect of Different Vegetable Oils on the Flavor of Fried Green Onion (Allium fistulosum L.) Oil"

_foods, 2023, doi:10.3390/foods12071442_

Round 1

Reviewer 1 Report

The article entitled "Effect of different vegetable oils on the flavor of fried green onion (Allium fistulosum L.) oil " presents an interesting and very useful work for fried green onion oil producers. Below I make a series of comments.

Lines 73-74: Indicate in each case whether the oil is refined, virgin, extra virgin. The denomination given by the authors is very general and can lead to misinterpretations.

Lines 179-180: The authors should have used robust statistics, and therefore based their sensory analysis conclusions on the median of the data, as this is more representative than the mean in the case of sensory analysis with tasters.

Figure 1: Replace F1 by PC1 and F2 by PC2.

Lines 204-205: What criteria have you used to make these two groups? Making an PCA with so few samples is not reliable. They should have studied more oil samples. For each vegetable variety they should have used at least three different oils.

Line 209: Indicate what the authors mean by “good flavor”

Author Response

Dear Editors and Reviewers:

Thank you for your letter and for the reviewers' comment concerning our manuscript entitled "Effect of different vegetable oils on the flavor of fried green onion (Allium fistulosum L.) oil. (Manuscript Number: foods-2239485) ". Those comments are all valuable and very helpful for revising and improving our paper, as well as the important guiding significance to researches. We have studied comments carefully and have made correction which we hope meet with approval. Revised portion are marked in the article. Please see the attachment.

Reviewer 2 Report

The manuscript described the flavor/aroma compound profile of seven vegetables after they have been used to fry green onions, typical of Chinese cuisine.  After frying, volatile compounds were detected, identified, and quantified by gas chromatography-mass spectrometry and electronic nose.  Several compounds were identified to contribute to the used oils’ flavor/odor. Overall, all the work is interesting and adds to existing literature on volatile carbonyl compounds from frying operations and frying chemistry.

Several areas need revision and improvement. The writing can be made more concise and several repetition should be deleted. Aspects through the manuscript should rewritten to enhance the meaning.

In the method, authors should have included the unfriend oils in the an analysis as a baseline to definitively establish flavour formation due to the frying process.

Specific areas to consider are listed below

Line 19: .... key aromas included ....

Line 22/23: Delete 'process' and 'evidently'

..... but the relative contents were not different. '

Line 24: ... frying oils .... heat degradation. ....

Lines 25/26: This should be revised. What kind of positive impact?

Line 32: Delete 'those'

Line 33: ... technique produces unique flavors at ....

Lines 36/39: .... properties of foods correlate with color.... palatability development [4]. ....... special flavors in fried foods ....

Lines 39-40: ---- a range of complex chemical reactions occur between the oil and the food material during frying ....

Lines 41/42: It should be noted that Maillard reaction is a browning reaction. Hence, authors should clarify what browning reaction is. Further, Maillard reaction is also associated with amino acid degradation. It is recommended to revise the section for clarity.

Line 43/44: ... The class of volatile compounds identified from deep-fried oils include ....

Line 46: Should be written in the present tense. 'Although comprehensive reports on .... green onion fried oils are lacking ......

What do authors mean by 'alliums generated'? Does the frying generate alliums or alliums are fried in the oils? Please clarify. This should be in the entire manuscript.

Lines 49/52: Consider this: 'Peng et al. [14] used headspace ...... and reported that .... contributed to grease flavor whiles ....... was associated with shallot scent flavor ....'

Lines 52/56: .... was used to compare ...... It was shown that sulfur-containing compounds ... significantly contributed to the ......

Line 54: Please clarify 'from microwave heat and gas heat treatment'. If products were fried, then why those treatments?

Was the analysis on the fried oil or shallot?

Lines 56-56: Consider this revision (does it capture the initial idea?): '... In heat treated garlic oils, volatiles varied due to the heat treatment or extraction methods used [16,17]. However, most of the detected volatiles were sulfur- or nitrogen-containing compounds.'

Lines 6168: The information is a bit unclear and repetitive. Please consider this revised and condensed version 'This study investigated the effect of different vegetable oils on the flavor profile generation of fried green onion oil. The fatty acid composition and volatile flavor compounds were determined and compared for seven fried green onion oils using sensory evaluation, GC-MS and electronic nose.'

Also, include the main outcome of the study. How will this impact food processing operations?

Lines 72/73: .... The vegetable oils, soybean oil (Yuanbo) ........

Line 76: What does 'newly produced' mean? Do you mean freshly processed? How was this determined?

Lines 82-96: Overall, the section has relevant information but needs to be improved/revised to make it succinct. There are repetitions especially the last two sentences. You might consider deleting those as they don't add much to the writeup.

Lines 89-90: Please clarify 'The minimum temperature in this process was slightly lower than 100 °C'.

Also, line 93: Please clarify 'above procedure was repeated twice....'. Do you frying was done multiple times? If so, consider stating this simply.

How much (weight) of the cut onion stalks was fried?

Lines 98-107: Consider: 'Fifty grams of oil was mixed with dichloromethane in a ratio of 1:4 (m:v). To this was added 1,2-dichlorobenzene (50 μL, 0.236 μg/μL) as internal standard and subjected to SAFE analysis to remove non volatile materials [18]. .........'

Line 101: What is a 'compact'? Maybe use 'glass compact'.

Lines 109: ...... performed on a TRACE 1310 ...... Samples were analyzed on a TG-Wax column ...

Line 124: .... of the detected compounds using equation 1, .........

Lines 126-127: Which authentic compounds? Are these different from the series of n-alkanes? If not, clarify in the chemicals section of the materials. If the same, then delete as it repeats and earlier point. Also, you can delete 'The results .... compound:'

Lines 128-131: Consider simplifying the statement. Do you need the carbon numbers in the calculation, or you only need the retention times? I suspect it's the later as the Kovat retention index is based on the retention times. Consider this 'where t'(n) and t'(n+1) are the retention times of the n-alkanes eluting before and after the compound "i", respectively, and t'(i) is the retention time of the compound (t'(n) < t'(i) < t'(n + 1) ).. '.

Lines 132: 'The relative concentration .......'

Lines 135-138: Consider:  ... where w(i) is the detected volatile compound; w(s) is the concentration (add unit) of the internal standard; and ........,

respectively. The relative correction factor .........'

Lines 140/141: Please correct this. The E-nose was not used to identify the oils - was this the case. It was used to analyze the oils for .......?

Line 142-148: Consider: '.... consisted of 10 metal oxide .... electronic nose software, Winmuster was used for ... and multivariate analysis. '

Line 149-153: Don't start sentence with the number digit. Spell it in full.

'Seven grams of each fried oil was put into 15 mL airtight vials (concentration chamber). The tubes were then heated at 100 °C for 30 min using a

SCILOGEX BlueSpin LED digital hot-150 plate magnetic stirrer (MS-H280-Pro, Berlin, CT, USA). Subsequently, a Luer-lock needle (20 gauge) connected to .... of the vials to draw the volatile gases in the headspace. Clean air was fed through a second needle connected to a charcoal filter to replace the air sampled from the headspace.... '

Line 153: should '20 g' be '20 gauge or '20 GA' so not to confuse with grams 'g'.

Line 162: Determination of fatty acid composition.

Please state the column details and the type of GC-MS (model).

Lines 162-163: Please clarify these statements. What is 'total oils' and 'isolated fractions of fried oils'?

Lines 164: FAME was analyzed in each sample using heptadecanoic acid as internal standard. The temperature programming and the GC-MS parameters were as previously described [22, 23].

Line 168-180: Please clarify 'relative humidity profile'

Please consider this: 'A descriptive sensory analysis was conducted at the Beijing Key Laboratory of Flavor Chemistry, Beijing Technology and Business University). Analysis was carried in a sensory laboratory maintained at 20±1 °C. 'Aroma profile analysis was completed by a panel of ten consisting of six females and four males, aged 23 to 30.... A 10 g sample ...... In addition to the descriptive analysis, panellist also scored the

aroma intensive using a linear scale ranging from 0 .... to 9 .... The final score for each odor attribute was the average from all panellists. Duplicate determinations were done for each sample.'

Line 182-187: This section needs to improve. Consider this example.

'Statistical analysis was conducted using the XLSTAT software v.2018. One way analysis of variance (ANOVA) was used to determine differences between samples. ..... Whilst the partial least square regression (PLS) ....'

Line 188: Should 'Conclusion' be 'Results'?

Line 191: Change 'consequence' to 'results.

Line 192: Is the descriptive aroma analysis not part of the sensory evaluation? Please clarify.

Lines 192-193: This statement could be made concise. E.g., 'The strongest flavour profile of the fried oils was the fried note, followed by ....'

Lines 194-196: Please clarify this statement or simplify. It is a bit unclear.

Line 196-197: Consider '... Compared to the other six oils, fried soybean oil (S1) had the best flavour profile.'

How did you assess soybean to have better flavor profile?

Line 199: .... Principal component analysis of seven fried green onion oils ....

Is seven green fried onion oil an accurate description? It suggests the oils came from the green onions.

Also check the accuracy of the caption.

Line 201: .... the sensory scores. The first two ....

Lines 210/211: Please clarify this 'and a significant role was played by the sensory evaluation in contrasting the differences of flavor'

Please which is PC1 and PC2 in the plot.

Lines 213-215: There is no need to repeat what was done as it's already in the methods.

Simplified: Consider 'The volatile compounds identified in the seven fried oils are given in Table 1. A total of 103 volatile compounds were identified and included 2 alkanes, 18 aldehydes, 16 alcohols, 12 ketones, 5 acids, 5 esters, 19 furans and furanones, and 19 sulfur-containing and 7 nitrogen-containing compounds....'

Again, please check if 'green onion oil' is accurate as it suggests oil from the green onion than oil used to fry the green onions.

Line 222: Throughout the manuscript, consider changing '7' to 'seven'.

Also delete 'kind'

Lines 223-225: Again, you don't need this since this is already in the methods. This section should focus on discussing the results only.

Consider: Figure 2 is a heat map showing the concentrations of the different volatiles identified in the seven fried oils. Furan and furanone compounds were the highest concentrations in the seven oils followed .......'

Lines 227/228: Where is the result for this? Is this referring to the furans and furanones only? Also how was 'richer and better' assessed?

Did authors analyze the unfried oils to ensure they do not contribute to the compounds detected?

Line 288: Delete 'to analysis'. Also, what is being combined with Table 1?

Line 231: Delete ', but the other specific compounds were different.'

Lines 231-232: This is unclear 'Besides, aldehydes and sulfur-containing compounds were also identified with the higher concentrations in the green onion fried oils. Please clarify.

Line 235:  ...green onion fried soybean oil....

Line 238: ... fried palm oil....

Line 239:  ... were the primary compounds ........ fried camellia oil ...'

Lines 239-244: This seems to repeat lines 233-235.

Since Table already has the 'A' labels, maybe consider removing them from the discussion.

Lines 233-244: Is it worth mentioning all these compounds considering they are contained in Table 1. Maybe, it's fine. However, why mention only soybean, palm, and camellia oils?

Lines 244-245: Please clarify this. Is it relevant to the discussion? Suggestion: Maybe start the section with that sentence/statement - it makes more meaning that way.

Table 1 caption: Volatile compounds detected in seven fried green onion oils and their concentrations.

Also, are the Cas# relevant? Are two decimal places needed?

Footnote 1: The concentrations of compounds were calculated using the internal standard method and the results is shown as the Mean±standard deviation.

Footnote 2: Please give the reference for the 'published retention index'. What is their relevance of the RIs in the Table or to the current analysis?

Importantly, the RIs are not discussed or explored in the discussion, meaning they are not pertinent to the analysis and can be omitted.

Line 249: Please revise the subtitle – ‘Electronic nose analysis of the fried oils'

Lines 250-253: Again, this section reads like methods and statistical analysis. Ideally you want to focus on the results. Or keep it simple. There is no need for those two references, 24 and 25.

Line 272: Change 'Besides' to 'Likewise'.

Lines 282-284: Overall, the electronic noise effectively differentiated the different fried oils .......'

Please explain 'the true or false fried oil' - this is the first mention of them.

Line 286: Figure 3 - Write 'PLS' in full.

Consider .... PLS explaining ...

Line 290: Fatty acid ....

Lines 291-295: Again, this is only repeating the methods which should be avoid.

Simply: 'Figure 4 shows the changes in the saturated fatty acids (SFA), unsaturated fatty acids (UFA), monounsaturated fatty acids 292 (MUFA) and polyunsaturated fatty acids (PUFA) contents in the seven friend oils.

Also, why report on UFA, then MUFA and PUFA. Maybe consider PUFA and PUFA and remove UFA.

Line 301: Figure 1: Please provide the y-axis label and unit

Lines 304-315: Section could be made more concise.

Lines 308-309: '.... camellia oil increased .... colza oil decreased. Fried olive oil had the most increase in the UFA content ...........PUFA were soybean oil .... Conversely, MUFA and PUFA decreased for sunflower oil and colza oil.

Lines 316-318: Please revise to improve meaning.

Also, do you need S1, S2 etc stated each time the oils are mentioned?

Lines 319-320:  ... Contrast analysis showed that during frying the composition of fatty acids of the oils influenced the flavor of fried oil....,

Line 322-325: '.... due to their double bonds resulting in ... further decomposed to volatile ...... and so on, contributing to the characteristic flavor profiles.... '

Line 325-326: Do you mean that frying causes dehydration and cycling? If so, please make this clearer

Lines 326328: Is this relevant to the current work? Onion is not a protein food. What is the protein content of green onion? It is about 0.3 g. Is this enough to justify this reaction mechanism?

Lines 328-330: This seems to be in the wrong place. The fact that oleic and linoleic acids were the two main UFA should be mentioned at the initial discussion of the fatty acids and not here.

Also, why fried, and oily notes?

Also revise the sentence structure.

Lines 332-335: Please delete. Conclusion should summarize the main findings as per the objectives and make recommendations, but not to restate the method.

Line 336: Did the work assess the flavor of green onions or the resulting oil after frying the onions?

...... the electronic nose was sensitive to ... differences in the aroma of the seven vegetables after frying of green onions......... and sulfur-containing compounds, such as (E, E)-2,4-decadienal, (E)-2-heptenal, hexanal, dimethyl trisulfide, methyl propenyl disulfide, were highly correlated with the flavor characteristic of the fried oils....... The fatty acid results showed that ......

Line 346: Delete 'and identified, especially according to the changes in UFA content. '

Lines 346-350: Please revise to improve the meaning.

Lines 350-352: '.... This work provides preliminary data for future research to probe the factors influencing the flavor and quality of fried green onion oil products during frying.... '

Author Response

Dear Editors and Reviewers:

Thank you for your letter and for the reviewers' comment concerning our manuscript entitled "Effect of different vegetable oils on the flavor of fried green onion (Allium fistulosum L.) oil. (Manuscript Number: foods-2239485) ". Those comments are all valuable and very helpful for revising and improving our paper, as well as the important guiding significance to researches. We have studied comments carefully and have made correction which we hope meet with approval. Revised portion are marked in the article. The responds to the reviewers' comments are in the attachment. Please see the attachment.

Round 2

Reviewer 1 Report

The authors have adequately answered my questions. The article is ready to be published